# Unanswered Questions and Hypotheses about Domestic Cat Behavior, Ecology, and the Cat–Human Relationship

**DOI:** 10.3390/ani11102823

**Published:** 2021-09-27

**Authors:** Dennis C. Turner

**Affiliations:** Institute for Applied Ethology and Animal Psychology, I.E.A.P./I.E.T., 8810 Horgen, ZH, Switzerland; dennis@turner-iet.ch

**Keywords:** cats behavior, human–cat interactions, personalities, breeds, socialization, cognition, predation, space utilization

## Abstract

**Simple Summary:**

Over the last three and a half decades, many studies have been published about cat behavior and cat–human interactions (many by the author and his former team of assistants and university students); these have been summarized in recent review articles. Although we have learned much about domestic cats and their relationships with people, there are still many unanswered questions of interest to scientists and the lay public. The author has briefly referred to the past results and formulated the unanswered questions (some as hypotheses), even suggesting possible ways to answer or test them. Although the author is now retired, his intent is to encourage young researchers beginning their academic careers to take up the torch and work on this fascinating companion animal species.

**Abstract:**

After recent publication of several reviews covering research results from the last 35 years of domestic cat studies, a number of important unanswered questions and hypotheses have arisen that could interest active researchers, especially those beginning their academic careers. Some sections of this paper concern methodologies that have yielded new insights and could provide more in the future; other sections concern findings and interpretations of those that need further testing. First, hypotheses arise from combining subjective (or psychological) assessments of cat and human personality traits and observational (ethological) studies of cat–human interactions: e.g., do owners with high attachment to their cats interact differently with them than owners with low attachment levels? New analytical methods of dyadic interaction observations open the door for testing further hypotheses. In particular, the Theme^®^ (Noldus bv, NL) program could be used to determine if there are differences between cat breeds in interaction patterns with people, which is not only of interest to owners but also therapists employing cats in their practices. Cat breed differences have been found using subjective ratings, but these need to be corroborated by direct observational data from the home setting and/or non-invasive colony observations, since ratings based on anthropomorphic projections might not be reliable. This should be done before searching for the genetic basis of such differences. Reliable information on breed differences is also needed before prescribing certain breeds for animal-assisted interventions. A model has predicted that the degree of socialization as a kitten affects cats’ responses to positive and negative experiences with unfamiliar humans and their formation of feline–human relationships later on. This needs to be tested in an ethically approved manner on cats of known socialization status and has enormous consequences for cat adoptions from animal shelters. Observations of human–cat interactions have yielded many correlations, which can be tested by non-invasive manipulations of human behavior in the home setting. Examples of these will be given and are of general interest to the cat-owning public. A review of first findings on social cognition in cats has resulted in further unanswered questions and hypotheses. Finally, two aspects of domestic cat ecology will be considered (effects on wildlife and space utilization), which are of great interest to the public and conservationists alike.

## 1. Introduction

In the course of preparing three somewhat different reviews about cat behavior, cat–human interactions, and the mechanics of such social interactions [1,2,3], the author realized that there are still many fascinating unanswered questions and untested hypotheses circulating which are of interest to both researchers and the lay public. Much of what is now published about these topics (see [1,2,3]) stemmed from the now retired author’s own former research group. A fourth review about cat cognition research [4] pointed out what (little) is known about social cognition in cats and yielded further open questions. The purpose of this article is to entice young researchers to take up the task of answering these questions and testing those hypotheses while developing their own academic careers. After very briefly citing the original research results, the open questions or untested hypotheses will be formulated and, whenever possible, ideas on how these might be approached while considering ethically the welfare of the cats and humans involved. Section titles indicate the general topic of interest, some concerning methodological issues and others concerning remaining knowledge gaps.

## 2. Combining Subjective Assessments with Observational Studies 

Anthrozoology is indeed an interdisciplinary field, but researchers tend to use the methods of their own field of endeavor and publish only therein. One finds more interdisciplinarity in the practice of animal-assisted intervention programs than in research methods or topics. James Serpell [5] was one of the first to suggest combining subjective interpretations of companion animal (dog) behavior (psychological assessments) with observational studies. Turner and his team used this approach for feline studies yielding new insights into the human–cat relationship ([6,7,8,9,10,11]). However, many of those insights were based on correlations and need further manipulative testing to discover causality. For example, based on positive and negative correlations between human-to-cat attachment levels, human and cat social support levels for the humans, and number of persons in the household, the authors [7] interpreted that for some people, cats may substitute for persons in the social network, but for most, they may be a source of additional support for people, particularly for those who are strongly attached to their companion cats. This is indeed an area requiring further research: On an independent sample of persons with equivalent human social support networks (qualitatively and quantitatively), do persons strongly attached to their cats receive more emotional support from them than those less strongly attached persons? However, the current level of need for social support of the persons involved will also have to be controlled for.

In another study [8], the owners of cats with outdoor access subjectively rated their animals as being less curious than the owners of indoor cats. The authors suspected that cats kept exclusively indoors might be compensating for their less animate environment by initiating more contact with objects (i.e., showing more curiosity). This needs to be tested. The frequency of investigating the same unfamiliar objects by indoor cats and cats with outdoor access when at home (per minute of presence in the home) could be compared as well as with the frequency of investigating similar unfamiliar objects when the outdoor cats are outside. In that same study [8], cats subjectively rated as being less predictable in their behavior were significantly more often rated high on aggressiveness toward their owners. If an owner is not good at identifying the cat’s emotional/motivational state, he or she is predicted to suffer more bites and scratches during interactions than an owner good at predicting the cat’s behavior. This requires more investigation while controlling for the amount of understanding of appropriate and inappropriate interactions with a cat, since that could also influence whether an owner is bitten or scratched.

More research is also needed combining subjective assessments of cat and human personality traits and observational (ethological) studies of cat–human interactions. This has proven useful in the past, but much remains to be discovered. Cat “individuality”, i.e., individual differences in behavior, is known to all cat guardians and seen in all cat studies (Turner [11]; Mendl and Harcourt [12]) and its origin, development, and stability need closer examination. Lowe and Bradshaw [13,14] have found that some of the effects of original socialization of kittens to humans remain stable for (at least) three years, e.g., being able to be stroked and held close. Feaver et al. [15] in Cambridge and Turner et al. [16] in Zurich asked persons familiar with the cats in each of their colonies to rank them for “friendliness to people”. In both of the colonies, the kittens were sired by two fathers and in both colonies, the “friendly” offspring were disproportionately sired by one of the two fathers. There were no other differences known between the two groups of kittens, e.g., in handling history or environment, which might have explained this result. The fathers were not in any way involved in raising (or even seeing!) their offspring, indicating a genetic paternity effect. Later, McCune [17]—using observational data—was able to show that this paternal effect on “friendliness” of the offspring was indeed on “boldness” with respect to exploratory behavior (see Turner [1] for more details).

Kotrschal et al. [18] and Wedl et al. [19] (see also Section 4 below) presented the most recent analyses of combined data from subjective ratings of cat personality traits and observational data on cat–human interactions. Cat personality was defined along four axes by the PCA of the subjective ratings of the observers, while human personality was defined with use of the NEO-Five Factor Inventory. Analysis of the observational data of cat–human interactions showed that dyads with male or female owners differed in only three of 2018 variables (1% of all behaviors), but it was the behavior of the cat, not of the owner, that resulted in those sex differences. Looking at the interplay between cat and owner personality, the authors found that the higher owners scored in Openness, the less anxious and tense were their cats and the more often these cats ignored the object in a novel object test. Those owners who scored high on Neuroticism viewed their cats as emotional supporters and probably offered their cats a less secure base than owners who scored high on Openness. Open owners viewed their cats more as play companions than as social supporters. Furthermore, cats of open owners relative to neurotic owners may develop into somewhat more secure and less anxious animals. This prediction remains to be tested during long-term observations in selected households.

## 3. Physiology and Human–Cat Interactions and Relationships

Although this is potentially a very broad field of investigation, the author will mention only one area of particular interest to researchers and practitioners alike: The role of oxytocin and other hormones in the establishment and maintenance of human–cat relationships. Although oxytocin has received much professional and public attention for human–human relationships, this has only recently taken place concerning human–animal interactions (Beetz et al. [20]), and much remains to be discovered. 

Beetz et al. [op cit] concluded in their review that the effects of oxytocin and human–animal interaction largely overlap and that the latter affects the oxytocin system. Rault et al. [21] found that most of the studies on oxytocin in domestic animals have been published fairly recently with the majority of those on dogs and only 3% on cats. One promising study by Arahori et al. [22] had cat owners subjectively rate their cats on some 30 questions, from which then a factor analysis identified four personality factors (Openness, Friendliness, Roughness and Neuroticism). Genetic analysis of the exon1 region of OXTR in DNA samples from those cats found three nucleotide polymorphisms. Those cats with a particular allele in one of the polymorphisms showed significantly higher roughness scores than cats without that allele. The authors proposed that genetic variation in cats might be linked to their personality traits, and this requires further research.

Using a relatively new method for quantifying the urinary concentration of oxytocin in cats, Nagawasa, Ohta, and Uchiyama [23] assessed the effects of social contact with humans on the levels of urinary oxytocin and cortisol metabolite in a relatively small sample of cats. Nevertheless, the results suggest that cats recognize social interaction with humans as important, and the study calls for a larger number of both cats and urine samples in future studies.

Kobayashi et al. [24] used functional near-infrared spectroscopy to assess the effects of touching and stroking a real or soft toy cat on the prefrontal cortex, particularly the inferior frontal gyrus (IFG), by male and female graduate students. The students also completed the Self-Assessment Manikin to measure their emotional responses and the NEO-Five Factor Inventory to assess their personalities. During the tactile interactions with the real cat, the values of oxygenated hemoglobin in the left IFG of the females were significantly greater than in the males and higher than after stroking the toy cat. Activation levels of the left IFG were also positively correlated with neuroticism when stroking the real cat. Touching/stroking the cat improved the mood of females, especially those with higher levels of neuroticism. The authors suggest that the physiological effects of interacting with a cat may be different between the sexes and this needs further investigation, given other sex differences found in human–cat interactions [1,2].

## 4. Application of Analytical Methods for Dyadic Interactions

Kotrschal et al. [18] and Wedl et al. [19] used relatively new analytical methods to study the structure of human–cat interactions observed in the home setting for the first time. Wedl et al. used Theme^®^ (Noldus bv, The Netherlands) to analyze strings of video-recorded owner and cat behaviors during four visits to 40 cat-owning households. As summarized by Turner [2], the Theme^®^ algorithm searches the data for sets of events that follow each other non-randomly in the temporal sequence. Two actions that occur repeatedly and regularly in alteration form a so-called “t-pattern”. Hierarchically structured t-patterns emerge from the detection of relationships of these previously detected patterns by repeated use of the algorithm scanning the strings of behaviors. Then, Wedl et al. [19] looked at factors that influence these temporal patterns and predicted that because most owners regard their cats as social companions, the dyadic structure would be contingent on owner and cat personalities, sex, and age, as well as duration of cohabitation of the partners. The authors found that both partners’ personalities and sex and cat age had significant effects on the t-patterning. They suspect an influence of housing conditions (indoor vs. outdoor access) on the t-patterns as found by Turner [9], but this remains to be tested. Further questions of interest would be: Is the t-patterning different in interactions with cats of different breeds? (See the following section.) Is it different in interactions with intact and neutered/spayed animals? Overall, this method of sequential data analysis between social partners has just begun to be tapped and shows great promise for the future. This is important in connection with communication between cats and their guardians and interpretation of behavior.

## 5. Differences between Cat Breeds and Relevance for Animal-Assisted Interventions

Hart and Hart [25] surveyed some 80 feline veterinarians who were considered to be unbiased authorities on breed differences in cats, randomly selected from across the USA. Each ranked a random selection of seven (five breeds plus domestic short- and long-haired cats) out of 15 cat breeds under consideration along 12 behavioral traits. The authors found that three of the traits had high predictive value to distinguish the breeds, seven had moderate, and two had low predictive value. Wilhelmy et al. [26] used a well-established questionnaire (the Fe-BARQ) to generate “standardized behavior profiles” of 12 breeds (also of interest, various coat color variants and eye color effects). Salonen et al. [27] used another health and behavior questionnaire completed by owners and determined behavioral differences between 19 cat breeds and breed groups along 10 different behavioral traits. A moderate level of heritability in three breeds and seven traits was found, but the authors reported that substantial genetic variation still existed within the breed populations. The three breeds were Ragdoll, Maine Coon, and Turkish Van, while the seven traits were activity level; contact to people; aggression to strangers; aggression to family members; shyness toward novel objects; and shyness toward strangers. Most of the aforementioned studies have used subjective ratings of behavioral traits, albeit one by cat experts or with large samples of cat owners and standardized, properly analyzed questionnaires; but to date, with the exception of Turner’s [10,28] studies, none has looked at observed behavioral differences to validate these subjective findings. This is indeed a wide-open research field with practical consequences for animal-assisted interventions involving cats. Practitioners who work with cats in AAI, e.g., Frick and Tanner-Frick [29], either tend to work with Siamese cats (known as “the dog” among cat breeds) or quiet, less boisterous breeds such as the relatively large Ragdoll or Norwegian Forest cats. Often, the author has been asked which cat (or dog, for that matter) breed is best for which client/patient group or ailment category. The answer is simply that we do not know (yet!).

## 6. The Effect of Later Experiences with Unfamiliar Humans on Well- and Non-Socialized Adult Cats 

Based on the results of several studies, Turner [28,30] proposed a model predicting differential outcomes of later positive and negative experiences with people depending on the quality of original socialization to humans as a kitten (see Figure 1).

Those factors that have been shown to influence the establishment of a first relationship (see Turner [1]) appear in the middle of the figure, including stroking the kitten during the sensitive period of socialization [31]. It has been hypothesized that a well-socialized cat can withstand many negative experiences with other people before becoming wary of such contacts and requires very few positive experiences with a new guardian to become friendly and trusting of that particular person (Turner [3]). On the other hand, a cat poorly socialized to people as a kitten needs many positive experiences to accept a new person, but few negative experiences with that person to confirm its wariness and fear of people. This has enormous welfare implications for cats in shelters in that poorly socialized cats take up limited space for longer while waiting for the personnel to find such a patient new owner, and well-socialized cats can be rehomed more easily and quickly. An ethically acceptable method to test the effects of negative (perhaps just lack of positive) and positive experiences with unfamiliar people needs to be developed and tested on cats of measurable socialization status (Kessler and Turner [32]) in order to test the model predictions. 

## 7. Social Interactions with Humans 

Turner’s ethological studies of contact initiation between cats and their owners (summarized in Turner [3]) have resulted in many significant “effects” of different parameters and conditions that have been loosely interpreted. For instance, the more the cat of the relationship initiates the contact, the greater the total interaction time between the owner and the cat in that relationship. In future research and after controlling for the usual amount of contact initiation by the owner, one could instruct the owner to experimentally increase (or decrease) his or her initiation to test if that indeed affects total interaction time in the relationship. Whether more contact time is indicative of a more harmonious relationship remains to be examined. Or, the author found that indoor cats initiate a higher proportion of the contacts with the owners than cats that are allowed outdoors do when at home and speculated that the indoor cats might be compensating for lower levels of environmental stimuli indoors with increased human contact. Not only could examination of novel objects change by temporarily restraining an outdoor cat’s access to the outdoors for a day or two (which might cause stress, indeed an additional important research question) but also the amount of contact initiation by the cat. However, a confounding factor would be that cats might be allowed outdoors because they are (or are perceived to be) less sociable. All of these points are important, since they could affect the human–feline relationship and/or cat welfare. 

Turner [9] found a significant positive correlation between the owner’s willingness to comply and the cat’s willingness to comply with the partner’s wishes to interact (all objectively defined) at other times. Again, correlations do not indicate causality, but this could easily be tested by experimentally instructing the owner not to comply, or to more frequently comply with the cat’s approaches and vocalizations. 

On a more practical level, men sometimes complain that household cats prefer to interact with the women in an abode (or even like them better). During public lectures, the author has often suggested that they feed the cats several days in a row (which often women regularly do), and/or to move down to the level of the animals when interacting (which women have been found to do, Mertens [33]) to elicit a change in the cats’ interactive behavior. These are but two simple manipulations, which have never been tested but show promise for improving the men’s relationships. Whether female cats tend to approach men and male cats approach woman in a standardized situation has never been investigated (again an unanswered question), but it is of general interest also to potential cat guardians.

The results of studies by Flegr et al. in Prague must be mentioned here as well. Even though many studies have indicated positive effects of companion animals on human health and well-being, the participants in most of those were usually aware of the purpose of the studies; when this was not the case, keeping pets, especially cats, and even more so, having been injured by pets, was negatively associated with a number of aspects of quality of life [34]. Cat scratches, relative to cat bites, were more often associated with unipolar depression in a cross-sectional study of humans (probably due to infection with Toxoplasma) [35]. Furthermore, a strong gender effect of toxoplasmosis has been found on the pleasantness attributed to cat urine odor [36]: While infected women rated cat urine odor as being less pleasant than noninfected women, infected men rated the same odor as being more pleasant than did noninfected men, the differences being statistically significant. The role of odors (both feline and human) should probably be considered in any future study of cat–human interactions examining differences between men and women, male and female cats.

## 8. Social Cognitive Abilities in Cats 

Social cognition in cats has just begun to awaken the interest of researchers, and Vitale and Udell [4] provided the first review of what we know and what we still need to investigate. They identified largely unexplored areas and suggested the following questions for future research: Do cats alter their social behaviors for communication with humans? Are there differences in the cognitive abilities of feral, shelter, and household cats? Do lifetime experiences, even training, influence cat cognition? The latter will also be of interest to disentangle such factors as age and breed that may influence the working and long-term memory of cats. Furthermore, new tests of perception abilities in cats should also include olfaction, not just vision, since olfaction plays an important role in the social lives of cats but probably also individual recognition of humans. Studies have found evidence that cats can distinguish between individual humans, and Saito et al. [37] demonstrated that they can distinguish between the voices of their owners and strangers. Although we now have good descriptions of cat vocalizations, we need more work on what they mean when they employ them in interactions with humans; here, the phonetic methods first used by Schötz et al. [38] might be helpful. 

Even though Miklosi et al. [39] had already shown differences between dogs and cats in their ability to use human pointing gestures, especially that cats lacked some components of attention-seeking behavior compared with dogs, Pongracz et al. [40] in Hungary demonstrated that cats were indeed able to read and follow human gaze for referential information. Galvan and Vonk [41] determined that cats were only moderately sensitive to human emotions indicated by postural and vocal cues, but particularly so when displayed by their owners as opposed to strangers. The latter implies that learning is probably involved. Rieger and Turner [42] found a tendency for cats to react to negative moods of their owners when close to them with more vocalizations and flank-rubbing, while Turner, Rieger, and Gygax [43] later confirmed that cats alleviated negative moods, comparable to the effect of a human partner.

Especially for high-intensity emotions, Quaranta et al. [44] demonstrated experimentally that cats are able to cross-modally match pictures of emotional faces with their related vocalizations in both conspecifics and humans. These authors proposed that cats have a general mental representation for the emotions of their social partners both conspecific and human, but that remains to be proven.

All of the above-mentioned results and unanswered questions have significant implications for understanding the feline–human relationship and, in many cases, improving the welfare of the cats in such relationships. 

## 9. The Ecology of Predation by Domesticated Cats 

Ever since Churher and Lawton [45] in 1987 through Loss, Will, and Marra [46] in 2013 and later, outdoor cats have been accused of eliminating wildlife, especially bird species but also small mammals, and reducing biodiversity. Although there is no doubt about the truth of this on small oceanic islands where cats have been introduced (and sometimes left behind) by humans and the potential prey species lacked defensive strategies having evolved in the absence of endemic predators (Fitzgerald [47]; Fitzgerald and Turner [48]), Lynn et al. [49] have questioned the “moral panic” over outdoor domestic cats destroying wildlife and reducing biodiversity. In a very recent review, Turner [50] has summarized what is known about cat predatory behavior, considered those facts in a fair appraisal, and explained why the results of such studies as those mentioned at the outset have been overrated and misinterpreted by many conservationists, wildlife biologists, and the media. In particular, although any number of studies have extrapolated the numbers of prey items carried home by the estimated number of cats roaming outside and arrived at enormous numbers (millions to billions), none have ever mentioned the estimated prey population size. When one considers this for the Loss et al. [46] study calculating 1.4 to 3.7 billion birds annually in the USA and juxtapositions this to the 20 billion birds breeding each year according to the US Fish and Wildlife Service, those cats are taking what one might expect for a normal predator–prey relationship. Furthermore, there has been only one long-term (3 year) study of cat predation at all life stages on a close-to-the-ground breeding songbird population in an area densely populated with cats [51]. The authors were able to calculate predation rates on all life stages of the birds and demonstrated that the cats did not push this songbird population into “sink” status. Concerning affects on biodiversity, it is important to realize that local effects, especially what we can see around our residential areas, only contribute to alpha-diversity, but what counts on the species level is gamma diversity—which is rarely mentioned (Turner [50]). Alpha diversity is measured very locally in individual habitats; beta diversity is a measure of the heterogeneity between habitats; while gamma diversity (or biodiversity) is the overall species diversity of a range of habitats or communities within a larger region [52]. There has now been sufficient research on methods to reduce cat predation on birds and small mammals (from temporary confinement and over supplemental feeding to small bells and colorful collars), but more studies are needed to (1) put predated prey numbers in relation to estimates of total prey population size; (2) consider the effects of cats on reptiles and amphibians (again, juxtaposed against prey population estimates in a larger region); and (3) measure changes in the productivity of local prey populations over longer periods as Weggler and Leu [51] did.

## 10. Space Utilization by Cats Allowed Outdoors

Two questions are often posed by cat owners and need further attention from the research community: What are the home ranges of household cats allowed outdoors, and how do they overlap with those of neighboring cats? Given the frequent anecdotal reports about cats finding their way home even over hundreds of kilometers, how good are the homing abilities of the domesticated cat?

Those field studies from around the world, which reported measurement data on the home ranges of cats, were included in an earlier review article [52,53] and used to mathematically relate home range size, cat population density, and prey/food abundance and distribution at the various study sites. Those studies were occasionally conducted with radio-collared individuals but before the advent of GPS collars with automatic tracking on a laptop, which have been shown on several popular TV documentaries. However, the author is unaware of any recent scientific paper analyzing such data in detail, which would be a welcome addition to our knowledge set. 

The same is true for the many anecdotal reports of cats traveling (even) hundreds of kilometers from some distant point to find their way home. It is plausible that through random wandering such a cat might come upon an area, which it recognizes or knows from its acoustic/olfactory character and then proceeds to its home base. However, the ancestor of the domestic cat was a stationary (territorial) species [54], not a migratory animal, and under no selection pressure to develop homing abilities. Although an interesting, unanswered question, ethical considerations probably prohibit any research to provide clarification.

## 11. Conclusions

Much has been learned over the past several decades about cat behavior, human–cat interactions and relationships, cognition, and ecology and has been presented in recent review articles. Interpretations of the results have not yet been tested! The author has pointed out many open questions that are begging to be answered and of interest to both researchers and cat enthusiasts. Of course, there are many more questions than presented here, and it is hoped that active (perhaps young) researchers will be encouraged to work with, and on this fascinating companion animal.

## Figures and Tables

**Figure 1 animals-11-02823-f001:**
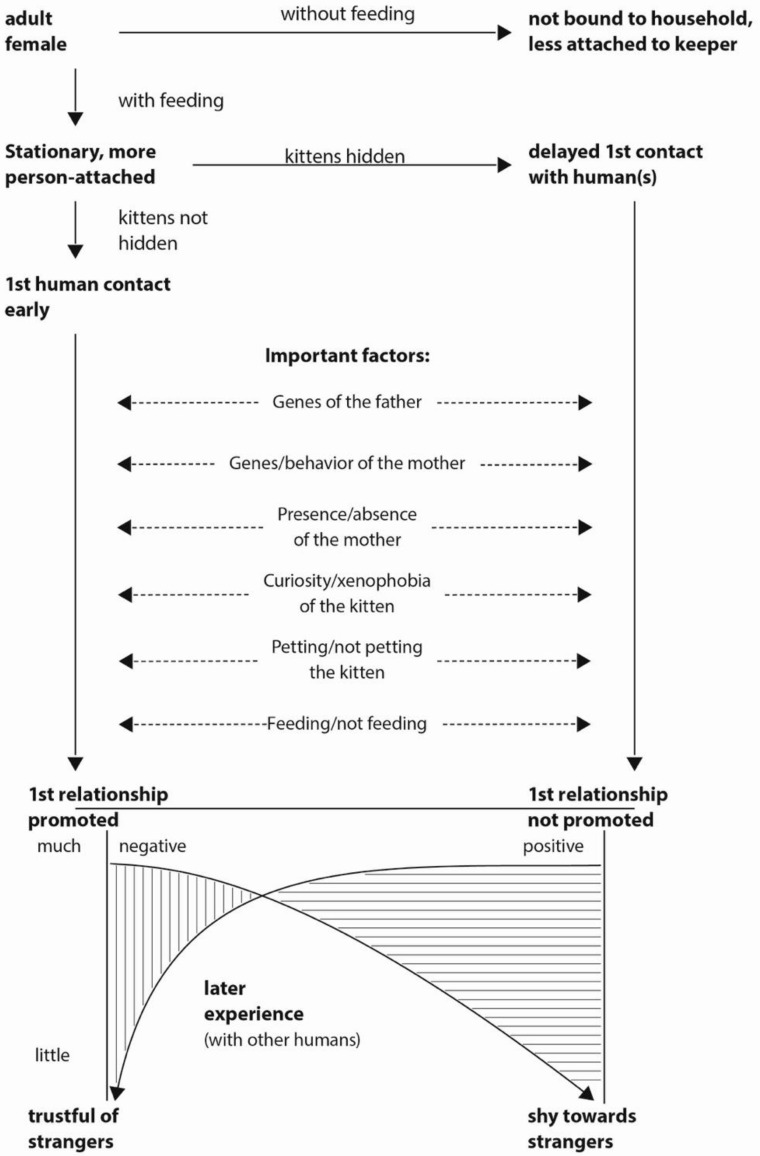
A model of the factors, which influence a kitten’s first relationship with a human, and of how later experience with other humans affects the cat. After Turner [30].

## Data Availability

Data over the decades were archived at the Zoology Institute (now the Dept. of Evolutionary Biology and Environmental Studes), University of Zurich-Irchel, Switzerland.

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
