# Peer review of "Unanswered Questions and Hypotheses about Domestic Cat Behavior, Ecology, and the Cat–Human Relationship"

_animals, 2021, doi:10.3390/ani11102823_

Round 1

Reviewer 1 Report

Thank you for writing this paper, as it has the potential to be of great value to early career researchers in feline behaviour. The article is equally a “mini” review of current knowledge and a “call to action” for new researchers, focusing specifically on interesting new research approaches, and research gaps in feline-human interactions (such as the influence of physiology, breed, socialisation, contact initiation, and feline cognitive abilities) and the impact of feline predation on ecology. It is clear that vast experience and insights from decades of feline behavioural research have been drawn upon to write this. Sections five, (“Differences between cat breeds and relevance for animal assisted interventions), six (“The effect of later experiences with unfamiliar humans on well- and non-socialized adult cats”), and nine (“The ecology of predation by domesticated cats”) are particularly interesting, as the relevance of the potential research to human and animal welfare is quite clear. However, to effectively target and be of most benefit to early career researchers, a number of aspects could be improved. At times, the article is difficult to follow. This may be due to its structure. Some sections focus on research methodologies (2. Combining subjective assessments with observational studies; 4. Application of analytical methods for dyadic interactions) and the remainder on research gaps and this is not immediately clear. It would be best to standardise this/make it more consistent. For example, section one could be changed into a “Feline personality/behavioural tendencies” section (with some material redistributed amongst other relevant sections) and section four could be fused with another related topic. In addition, the research gaps and/or hypotheses need to be explicitly and clearly stated so that they “stand out” and this is not always the case. I also recommend including the potential significance/applications of all the proposed new study areas (explain why they deserve attention), in order to inspire researchers to “take up the mantle”. Although not essential, it would be interesting to read about possible barriers to the proposed research. For example, challenges in project design and/or logistics due to multiple confounding factors? Or specific ethical issues? Finally, the article should stand on its own and not require prior knowledge of the review articles it is based on. Therefore, some points require further explanation. Please see more specific comments below. I enjoyed your article and sincerely hope that my comments are helpful.

Lines 19-21 (abstract): “First hypotheses arise from combining subjective (or psychological) assessments of cat and human personality traits and observational (ethological) studies of cat-human interactions.”

Comment: Such as? Provide a few additional details.

Lines 21-22 (abstract): “New analytical methods of dyadic interaction observations open the door for testing further hypotheses.”

Comment: Such as? Provide a few additional details.

Line 23 (abstract): Suggest changing “subjective assessments” to “subjective ratings”.

Lines 24-26 (abstract): “This should be done before searching for the genetic basis of such differences and before prescribing certain breeds for animal-assisted interventions.”

Comment: Add a rationale for this.

Lines 26-28 (abstract): Suggest rewording to “A model has predicted that the degree of socialisation affects cats’ responses to positive and negative experiences with unfamiliar humans and their formation of feline-human relationships” or similar

Line 28 (abstract): Change “manor” to “manner”.

Lines 30-32: “Observations of human-cat interactions have yield many correlations, which can be tested by non-invasive manipulations of human behavior in the home setting. Finally, a review of first findings on social cognition in cats has resulted in further unanswered questions and hypotheses.”

Comments: What correlations, hypotheses, and unanswered questions? Why is this an important area for research? Provide a short overview.

Lines 40-41: Add citations and remove “(references follow below)”

Lines 54-58:James Serpell [5] was one of the first to suggest combining subjective inter- pretations of companion animal (dog) behavior (psychological assessments) with observational studies and Turner and his team followed up in their cat studies yielding new  insights into the human cat relationship (Bahlig-Pieren & Turner, [6]; Stammbach & 57 Turner, [7]; Turner & Stammbach-Geering, [8]; Turner, [9]; Turner, [10]; Turner, [11]).”

Comment: I think that Stammbach & Turner (1999) only used questionnaires with no observational methods. However, these were objective not subjective tools.

Line 56: Suggest changing “Turner and his team followed up in their cat studies” to “Turner and his team used this approach for feline studies”

Lines 60-65: Suggest reworking, as this sentence is too long.

Lines 65-68: “On an independent sample of persons with equivalent human social support networks (qualitatively and quantitatively), do persons strongly attached to their cats receive more emotional support from them than those less strongly attached persons?”

Comments: Suggest making this sentence stronger to clarify that this is a research gap. Include why this is an important area for study. Also, in one of your other cited reviews (Turner, 2017), you stated that cats help to improve a “depressive” mood in humans but do not affect positive mood. Therefore, it is likely that people will only need emotional support when going through challenging times. Some may experience more challenging times than others over a period of time and therefore need or receive more emotional support from their cats during that period. This would need to be controlled for.

Lines 72-75: Clarify that this is a research gap, as above and why this is important/significant. Also, it would be necessary to standardise the time that both indoor and outdoor cats have available for interaction with the objects.

Line 75: “In that same study”- repeat citation.

Lines 75-79: “In that same study, cats subjectively rated as being less predictable in their behavior were significantly more often rated high on aggressiveness towards their owners. If an owner is not good at predicting the cat’s next move he or she is predicted suffer more bites and scratches during interactions than an owner good at predicting the cat’s behavior.”

Comments: Suggest changing “predicting the cat’s next move” to “identifying the cat’s emotional/motivational state”. Make it clear that this is a recommended area for future research and why this is a significant area. It is also important to note that a lack of understanding of appropriate and inappropriate interactions with a cat can also influence whether an owner is bitten or scratched.

Lines 80-81: “More research is also needed combining subjective assessments of cat and human personality traits and observational (ethological) studies of cat-human interactions.”

Comments: Why does this deserve further attention? Why is it important to identify and describe feline personality? What feline personality traits have already been characterised?

Lines 81-83: “Cat 81 “individuality”, i.e., individual differences in behavior and personality traits, is known to all cat guardians and seen in all cat studies (Turner [11]; Mendl and Harcourt [12])..”

Comment: Cat personality is not explored in all feline research projects, therefore it would not be correct to state that it is “seen in all cat studies”.

Lines 85-86: “some of the effects of original socialization of kittens to humans remain stable for (at least) three years.”

Comment: What specific effects of socialisation?

Lines 90-91: “The fathers were not in any way involved in raising (or even seeing!) their off- spring indicating a genetic paternity effect.”

Comment: Could any other factors be involved other than a paternal genetic effect? The environment, handling etc., for example?

Lines 111-114: “Although this is potentially a very broad field of investigation, the author will mention only one area of particular interest to researchers and practitioners alike: The role of oxytocin and other hormones in the establishment and maintenance of human-cat relationships.”

Comment: Why exactly is oxytocin of particular interest?

Lines 116-118: “Rault et al. [21] found that most of the studies on oxytocin in domestic animals have been published fairly recently with the majority of those on dogs and only 3% on cats.”

Comment: Is Rault et al. the correct citation? I could not find this information in their article.

Lines 136-136: “During the tactile interactions with the real cat, the values of oxygenated hemoglobin in the left IFG were significantly greater than in the males and higher than after stroking the toy cat.”

Comment: significantly greater in whom than in the males? Significantly greater in females than in males?

Lines 141-142: “The authors suggest that the physiological effects of interacting with a cat may be different between the genders and this needs further investigation.”

Comment: Why would this be important to investigate?

Lines 159-163: “ Further questions of interest would be: Is the t-patterning different in interactions with cats of difference breeds? (See the following section.) Is it different in interactions with intact and neutered/spayed animals? Overall, this method of sequential data analysis between social partners has just begun to be tapped and shows great promise for the future.”

Comment: Why does this warrant attention?

Lines 170-171: “Wilhelmy et al. [26] used a well-established questionnaire to generate “standardized behavior profiles” of 12 breeds (also of interest various coat color variants and eye color effects).”

Comment: What well established questionnaire?

Lines 174-176: “A moderate level of heritability in three breeds and seven traits was found but the authors reported that substantial genetic variation still existed within the breed populations.”

Comment: More details needed. What traits?

Line 210: “The initiation of social interactions”

Comment: Suggest changing title to “Social interactions with humans” or similar.

Lines 215-217: “After controlling for the normal amount of contact initiation by the owner, one could instruct the owner to experimentally increase (or decrease) his or her initiation to test if that indeed affects total interaction time in the relationship.”

Comments: Clarify that this a proposed area for future research. Why is total interaction time important? One could argue that the quality of the interaction may be as important or even more important than interaction time.

Lines 218-221: “Or, the author found that indoor cats initiate a higher proportion of the contacts with the owners than cats that are allowed outdoors do when at home and speculated that the indoor cats might be compensating for the lower levels of environmental stimuli indoors with increased human contact.”

Comments: Clarify that this a proposed area for future research and why it is important. Confounding factor: cats may be outdoor because they are (or are perceived to be) less sociable.

Lines 221-223: “Not only could examination of novel objects change by temporarily restraining an outdoor cat’s access to the outdoors for a few days (see above), but also the amount of contact initiation by the cat.”

Comments: This could cause high levels of stress (and therefore ethical issues), possibly resulting in withdrawal or aggression. Clarify that this a proposed area for research.

Lines 224-235: Why are these areas important? Could these things affect the human-feline relationship and/or animal welfare?

Lines 240-241: “Are there cognitive differences in the abilities of feral, shelter and household cats?”

Comment: Reword “cognitive differences in the abilities” to “differences in the cognitive abilities”.

Line 245: Citation needed.

Lines 247-248: “Studies have found evidence that cats can distinguish between individual humans..”

Comment: Citation needed.

Line 255: Change “attention getting” to “attention seeking”.

Line 264: Change “capable” to “able”.

Section 8 (“Social cognitive abilities in cats”) Lines 246-268: tie research review in with future research questions and explain significance.

Line 270: Consider rewording.

Line 288: Change “live stages” to “life stages”.

Lines 293 and 294: What does alpha diversity and gamma diversity mean?

Line 303: Change “several years” to “number of years” or similar.

Lines requiring paraphrasing (as they are currently too similar to writing in the author’s other works):

Lines 63-65 (Similar to writing in Turner, 2017): “but for most they appear to be an additional source of emotional support, especially for those people who are strongly attached to their animals”.

Lines 104-107 (Similar to writing in Turner, 2017): “Owners high in Neuroticism turned to their cats mainly as emotional supporters and hence, they might have offered a less secure base for the cat than the owners high in Open-ness. The later persons considered their cats companions for play rather than social supporters.”

Lines 198-203 (Similar to writing in Turner, 2021): “For a cat well-socialized to humans as a kitten  it is hypothesized that it takes many negative experiences with other people to become wary of such contacts and very few positive experiences with a new owner to become  friendly and trusting of that particular person (Turner [3]). A cat poorly socialized to people as a kitten requires a great deal of positive experience to accept a new person, but very little negative experience with a person to confirm its wariness and fear of people.”

Lines 252-259 (Similar to writing in Turner, 2021): Pongracz and his colleagues in Hungary have been particularly active in the social cognition area. Even though Miklosi et al. [36] had already shown differences between dogs and cats in their ability to use human pointing gestures, especially that cats lacked some components of attention-getting behavior compared with dogs, Pongracz et al. [37] demonstrated that cats were indeed able to read and follow human gaze for referential information. Galvan and Vonk [38] found that cats were only moderately sensitive to emotions as indicated by human postural and vocal cues, but particularly when displayed by their owners.

Lines 264-268 (Similar to writing in Turner, 2021): “Quaranta et al. [41] demonstrated experimentally that cats are indeed capable to cross—modally match pictures of emotional faces with their related vocalizations in both conspecifics and humans, especially for high intensity emotions. These authors concluded  that cats have a general mental representation for the emotions of their social partners  both conspecific and human.”

Author Response

Thank you for your very valuable comments and suggestions. Most I have accepted and made the necessary changes in tracked changes, with exception of one of the suggested changes in the Abstract. If I had given "examples" there ("such as, provide additional details") those are actually the main content of my article and would have made my abstract even longer. I hope you and the handling editor will accept my decision on this. Further, to rearrange and combine the various sections in the original manuscript would have required a COMPLETE re-write of the text and literature reference numbering, which, given the submission deadline for the revised manuscript, would be impossible for me to complete - and that with minimal improvement in the flow of the text. Reviewers 3  (2 and 4) were satisfied with the organization of the text. Nevertheless, I have made many of the recommendations you suggest in your general comments: In the Abstract I have also mentioned that some of the sections concern research methodologies and others research/knowledge gaps. I have more clearly stated those gaps and/or hypotheses so that they "stand out" for the reader. I have added a few statements about the significance/ applications of the proposed new study areas but NOT attempted summaries of the previous, published review articles, the purpose of those articles! However I have made sure that the reader understands what the unanswered question or research gap is. I have not mentioned the changes below, but made them all and these are shown in tracked changes in the attached, revised manuscript, which also includes changes and additions requested by Reviewer 3. Reviewers 2 and 4 required NO changes and recommended acceptance as it stood.

Reviewer 2 Report

Very interesting manuscript that will be of value to the research community, and provide a rich source of ideas for potential researchers in this discipline.

Abstract - "Manor" should read "manner"

Author Response

Thank you for your kind comments. I'm happy you appreciated my efforts.

Reviewer 3 Report

The author reviews what he has done and what he has not managed to investigate during his fruitful career with respect to the cat behaviour research. He summarizes the open questions and shares them with others. This is a very useful and likeable attitude and should become a more common practice in the scientific community. I can hardly criticize anything as the manuscript is actually a literature review based on published and peer-reviewed papers and it only points out the aspects worth of further investigation. The article is well organised and well written and I have only few minor comments.

Title: I think that ": an open field for young researchers" can be deleted without any harm. The title would be thus shorter and more straightforward. For sure not only young researchers are addressed.

Line 9: ... many from the author... I think that it should read ...many by the author

L 28 manor on cats should read manner on cats

L 51 should read Anthrozoology

L 57-58, L 83, L 93 and still on some few other places in the text: names of the authors are redundant here, leave only number of reference in square brackets.

L101, 142, 156 and maybe still elsewhere in the text: substitute gender with sex, definitely in all cases when you are speaking about animals. Currently, the word "gender" is much discussed in politics and society, but note please that there may be many genders but there are only two sexes, male and female. Gender is a word referring to psychology and sociology not to biology.

L118 should reas 3 % (i.e. a space between number and unit)

L122 Thoss cats should read Those cats

L189 - delete "later in German", this is clear from the list of literature (besides that: should read latter)

L229-234. My own observation is that (like dogs) female cats tend to interact more with men while male cats like more women. This becomes apparent when unfamiliar guests visit a household with cats. Is it only my anecdotic observation or has it been mentioned also in the literature?

I think that also some papers by J. Flegr et al. are relevant to be discussed here:  

DI: 10.1186/s13071-015-1290-7

DOI: 10.1016/j.schres.2017.02.007

DOI: 10.1371/journal.pntd.0001389

I have been a bit dissapointed to have found only one cahpter (chapter 9) to be devoted (to only one aspect of) ecology. There is an interesting BBC film "The secret life of cats" <https://www.dailymotion.com/video/x10w49m> about village cats equipped with GPS collars and action cams revealing nocturnal activity of cats, about which their owners had no idea. I have never seen any scientific paper examining this topic. Also there are anecdotic reports about homing achievements of cats which, after they had got lost, returned homeacross hundreds of km

 <https://frontpagemeews.com/category/cats/cat-behavior/homing-lost-cats/>

Has anyone studied homing and navigation abilities in cats systematically?

Author Response

Thank you for your very valuable comments and suggestions. I have accepted and made ALL of them, including adding the reference to Flegr's work and adding a 10th section on space utilisation (and homing) to answer your questions there. Please note that Reviewer 1 made MANY good comments that are now also included in tracked changes in the attached revised manuscript. I have included my comments on his/her review for your information here:

Thank you for your very valuable comments and suggestions. Most I have accepted and made the necessary changes in tracked changes, with exception of one of the suggested changes in the Abstract. If I had given "examples" there ("such as, provide additional details") those are actually the main content of my article and would have made my abstract even longer. I hope you and the handling editor will accept my decision on this. Further, to rearrange and combine the various sections in the original manuscript would have required a COMPLETE re-write of the text and literature reference numbering, which, given the submission deadline for the revised manuscript, would be impossible for me to complete - and that with minimal improvement in the flow of the text. Reviewers 3  (2 and 4) were satisfied with the organization of the text. Nevertheless, I have made many of the recommendations you suggest in your general comments: In the Abstract I have also mentioned that some of the sections concern research methodologies and others research/knowledge gaps. I have more clearly stated those gaps and/or hypotheses so that they "stand out" for the reader. I have added a few statements about the significance/ applications of the proposed new study areas but NOT attempted summaries of the previous, published review articles, the purpose of those articles! However I have made sure that the reader understands what the unanswered question or research gap is. I have not mentioned the changes below, but made them all and these are shown in tracked changes in the attached, revised manuscript, which also includes changes and additions requested by Reviewer 3. Reviewers 2 and 4 required NO changes and recommended acceptance as it stood.

Reviewer 4 Report

This review presents the state of the art of studies on some important aspects of cat behavior. The paper is the result of the author's great experience who knows how to focus, in a clear and synthetic way, on the knowledge already acquired and the problems that are still open. In my opinion this review is very important for the development of new lines of research that can clarify some fundamental aspects of feline behavior.

Author Response

Thank you for your kind comments. I'm happy you liked my manuscript.

Round 2

Reviewer 1 Report

Thank you for your comprehensive response. I note that you have adopted the majority of the additions/revisions I suggested. And I understand your reasons for not restructuring the paper. It is unusual for an eminent researcher to directly group together and share the insights and further ideas sparked during their own career and from their own work. Your paper will be a valuable addition to the literature.